# Gaining Insight from Semi-Variograms into Machine Learning Performance of Rock Domains at a Copper Mine

**Narmandakh Sarantsatsral * and Rajive Ganguli** 

Department of Mining Engineering, University of Utah, Salt Lake City, UT 84112, USA
* Correspondence: n.sarantsatsral@utah.edu

**Abstract:** Machine learning (ML) is increasingly being leveraged by the mining industry to understand how rock properties vary at a mine site. In previously published work, the rock type, granodiorite, was predicted with high accuracy by the random forest (RF) ML method at the Erdenet copper mine in Mongolia. As a result of the optimistic results (86% overall success rate), this paper extended the research to determine if ML would be successful in modeling rock domains. Rock domains are groups of rocks that occur together. There were two additional goals. One was to determine if the variograms could predict or help understand how ML methods would perform on the data. The second was to determine if 2D modeling would perform well given the disseminated nature of the deposit. ML methods, multilayer perceptron (MLP), k-nearest neighborhood (KNN) and RF, were applied to model six rock domains, D0–D5, in 2D and 3D. Modeling performance was poor in 2D. Prediction performance accuracy was high in 3D for the domains D1 (92–94%), D2 (94–96%) and D4 (85–98%). Note that the domains D1 and D2 together constituted about 80% of the samples. Conclusions drawn in this paper are based on the results of 3D modeling since 2D modeling performance was poor. Prediction performance appeared to depend on two factors. It was better for a domain when the domain was not a minuscule proportion of the sample. It was also better for domains whose indicator semi-variogram (ISV) range was high. For example, though D4 only contributed 15% of the samples, the range was high. MLP did not perform as well as KNN and RF, with RF performing the best. The hyperparameters of KNN and RF suggested that performance was best when only a small number of samples were used to make a prediction. One overall summary conclusion is that the two most important domains, D1 and D2, could be predicted with high accuracy using ML. The second summary conclusion is that semi-variograms can provide insight into ML performance.

**Keywords:** artificial intelligence; multilayer perceptron; neural network; random forest; k-nearest neighborhood; mining geology; mining industry

## 1. Introduction

ML has the potential to increase efficiency in mine planning by automating parts of mine planning. Traditional rock-type modeling requires frequent human intervention and may require significant re-modeling when new data are created. An operating mine generates new data every day when hundreds of blastholes are sampled for grade control purposes. Additionally, periodic exploration drilling creates new data. ML, once set up, makes it easy to update models for short-term and long-term planning purposes.

As part of the overall trend in the industry, the ML method, random forests (RF), was applied in a previously published work at Erdenet Mining Corporation's large Erdenet copper surface mine (EMC) to predict a particular granodiorite rock [1]. This was different from typical ML applications in orebody modeling, as most applications in the literature focus on grade estimation [2–5]. Predicting rock type is important as rock type affects mine operations. At EMC, RF performed well (86% overall success rate) when rock type was predicted two benches below the lowest bench being mined. This performance is

important, and relevant, for short-term mine planning purposes. The performance was even better (89% overall success rate) when predicted locations were close to existing data. This performance was relevant to everyday grade control where rock type is known in many locations in a drill block, and rock type has to be predicted just a short distance away from the blastholes. Given the good results, the research was continued in this paper to examine if ML could be successful in predicting entire rock domains, and not just one rock type. A rock domain is a group of related rock types that occur together. Modeling of rock domains is logical in porphyry copper deposits as the host rock and orebody contain various zones due to the complex nature of their formation [6]. The characteristics of supergene (near surface) ore minerals may be quite different from hypogene (deep) ore minerals. From an engineering perspective, rock domains are interesting to mines as they may have very consistent geotechnical or mineral processing characteristics. These characteristics affect the economics of mining operations, and thus, effective rock domain modeling can be a useful tool for planning engineers.

This paper contributes to the literature in two ways. Since the ML is applied to data from an operating mine to help geologists and engineers, the most obvious contribution is that the paper demonstrates the utility of ML in helping mine operations. The second contribution is explaining ML performance through the lens of semi-variograms. Semi-variograms are very insightful about the problem being studied. For example, the authors of [7] found parameters of variograms developed for forest images to be very important in understanding forest changes. In an academic exercise, a group of researchers demonstrated the link between sampling scheme, variogram parameters and kriging variance [8]. Variogram models were also determined to be important in the kriging and design of experiment-based modeling of aircraft wing shapes [9]. In a simplistic exercise, the authors of [10] demonstrate the effect of variogram sill and range on the estimate and estimation error. Thus, it would be reasonable to expect variograms to be insightful about ML performance. However, this has not yet been done. Some researchers have compared the performance of ML methods to geostatistical methods [11,12], but none have related how insights from variograms can be precursors to ML performance. In that, this paper is a first. Since variogram modeling is a common practice among orebody modelers, it would be useful to provide them with some guidance as to what could be expected in terms of ML performance given a particular type of variogram. It is hoped that this paper will build a bridge between the two different types of techniques, the somewhat black box ML methods and traditional variograms, so that traditional practitioners may anticipate what could be achieved should they opt for ML methods.

## 2. Geology and Drill Hole Data

The Erdenet copper surface mine in Mongolia mines the Erdenetiin Ovoo copper porphyry deposit. The deposit, a large copper-molybdenum deposit, occurs within the Orkhon-Selenge intrusive trough. The mine is located about 350 km from Ulaanbaatar, the capital city. The mine contains two deposits. The mining area is split into the Central (CEN) and Northwest (NW) areas. Initially created as one, the deposit was subsequently split by a fault. The CEN area got lifted during this geological process and moved along the fault to the south. The NW area stayed in place. NW and CEN are currently being mined and are, thus, the focus of this paper. The other two deposits, Shand and Oyut, are being explored.

The Erdenet mine has digitized its drillhole database recently. The drillhole database used in this research consisted of 2820 exploration drill holes, composited in 5 m lithological intervals. There were 84,062 lithological intervals between all the holes, of which 18,022 were in the CEN deposit and 66,040 were in the NW deposit. Besides the coordinates, the data included information on metals grades and rock types.

Geologists have classified the rocks in the lithological intervals into 21 types, including "unknown" (rocks that have not been characterized for various reasons). These rocks were then grouped into six domains, D0–D5 (Table 1). The rock characteristics, hardness,

density and strength are from [13]. Rock characteristics are only available for the major rock domains.

**Table 1.** The rock domains and their characteristics.

| No. | Group | Rock Types | Hardness, in Moss Scale | Density, (g/cm$^3$) | Strength, (N/mm$^2$) |
|---|---|---|---|---|---|
| 1 | Domain—D0 (ANDS) | (ANDS) Andesite | 7 | 2.11–2.36 | 225 |
| 2 | | (MONZ) Monzonite | 6–7 | 2.90–2.91 | 310 |
| 3 | | (DIOR) Diorite | 6–7 | 2.80–3.00 | 225 |
| 4 | Domain—D1 (BGDP) | (BGDP) Biotite granodiorite porphyry | 6–7 | - | - |
| 5 | | (GDRP) Granodiorite—porphyry | - | - | - |
| 6 | | (BGDI) Non-phyric biotite granodiorite | 6 | 2.60–2.80 | 175 |
| 7 | | (GRAN) Granite | 6 | 2.65–2.67 | 175 |
| 8 | Domain—D2 (GDIR) | (GDIR) Granodiorite | 6 | 2.60–2.80 | 175 |
| 9 | | (METS) Main mineralization stage metasomatic altered units predominantly granodioritic origin. | 6 | 2.60–2.80 | 175 |
| 10 | | (PLPO) Plagioclase—porphyry | 6 | 2.62–2.76 | - |
| 11 | Domain—D3 (Faults/Dykes) | (FLTZ) Fault zone | - | - | - |
| 12 | | (BRXX) Breccia | - | - | - |
| 13 | | (QPLD) Quartz plagioclase dike/dyke | - | - | - |
| 14 | | (DAPD) . . . dike | - | - | - |
| 15 | | (BPLD) Biotite plagioclase phyric dike/dyke | - | - | - |
| 16 | Domain—D4 (UNKN) | (UNKN) Unknown (initially has no lithology log) | - | - | - |
| 17 | Domain—D5 (Waste/Overburden) | SAPT—Saprolite | - | - | - |
| 18 | | GSPO—Grano-syenite porphyry | - | - | - |
| 19 | | DACT—Dacite | - | - | - |
| 20 | | RHYL—Rhyolite | - | - | - |
| 21 | | SYEN—Syenite | - | - | - |

Group D0 consists of all the harder rocks. The two most important domains are D1 and D2, as they contain a high proportion of metals. They contain biotite granodiorite porphyry and granodiorite, respectively. Together, they constitute 77% of the rock, with D1 making up 35% of the rock and D2 making up 42%. D4 makes up 15% of the rock and therefore, D0, D3 and D5 make up just 8% between all three domains. Table 2 shows the rock domain distribution in the CEN and NW deposits. The CEN deposit is clearly smaller in terms of exploration data. A difference between CEN and NW is that the domain D4 occurs in a much higher proportion in CEN (36.2%) than in NW (9.2%) deposit. Note that the percentages are rounded figures and so may not add up to 100%.

**Table 2.** Rock domain distribution in the two deposits.

| Domain | CEN | | NW | | Overall | |
|---|---|---|---|---|---|---|
| | Counts | Proportion | Counts | Proportion | Counts | Proportion |
| D0 | 417 | 2.3% | 3113 | 4.7% | 3530 | 4.2% |
| D1 | 4511 | 25.0% | 24,552 | 37.2% | 29,063 | 34.6% |
| D2 | 6323 | 35.1% | 29,172 | 44.15% | 35,495 | 42.2% |
| D3 | 97 | 0.5% | 2149 | 3.3% | 2246 | 2.7% |
| D4 | 6515 | 36.2% | 6052 | 9.2% | 12,567 | 14.9% |
| D5 | 159 | 0.9% | 1002 | 1.15% | 1161 | 1.4% |
| Total | 18,022 | 100% | 66,040 | 100% | 84,062 | 100% |

### 3. Methodology: Pre-Modeling Tasks and Modeling Strategies

In this paper, rock types were modeled with sample coordinates as inputs and rock domains as outputs. Modeling, using ML methods, was done in two dimensions (2D plan view) and three dimensions (3D). Since porphyry deposits are disseminated ore bodies there was an intellectual curiosity to determine if 2D modeling would be adequate or even outperform 3D modeling. Variogram modeling was also done to examine if their structure contained insights into the observed ML performance. However, to improve readability, they are presented after ML modeling performance is discussed.

ML methods require data preparation and modeling strategies. 2D modeling was particularly complex. Therefore, the methodology is presented across two consecutive sections. This section presents the pre-modeling tasks and the strategy, while the next section presents the specifics of each model and the results.

#### 3.1. Data Preparation for ML

The coordinates were normalized in the range $[-1, 1]$ prior to ML to avoid issues related to magnitude. The rock type, which was the output in the models, was encoded in a multiclass system as shown in Table 3. The multiclass encoding allows ML to handle all rock domains at once. As an example of the system, D1 would be represented as the vector $[0, 1, 0, 0, 0, 0]$.

**Table 3.** Multiclass encoding of the output.

| Domain Name | Multiclass Encoding | | | | | |
|:---:|:---:|:---:|:---:|:---:|:---:|:---:|
| D0 | 1 | 0 | 0 | 0 | 0 | 0 |
| D1 | 0 | 1 | 0 | 0 | 0 | 0 |
| D2 | 0 | 0 | 1 | 0 | 0 | 0 |
| D3 | 0 | 0 | 0 | 1 | 0 | 0 |
| D4 | 0 | 0 | 0 | 0 | 1 | 0 |
| D5 | 0 | 0 | 0 | 0 | 0 | 1 |

The data were subdivided for modeling and testing using a 75/25 modeling/testing split. In modeling, it is important to show that the inputs and outputs of the testing subset are similar to the inputs and outputs of the modeling subset. Therefore, Tables 4 and 5 show the attributes of each split for 2D modeling cases for NW and CEN deposits, respectively. The tables show that the inputs (x and y coordinates) are similar in the two subsets across several percentiles. These percentiles are distributed throughout the data and therefore, represent the data well regardless of skewness. Thus, comparing the two subsets at these percentiles demonstrates that the two subsets are similar. The distribution of the outputs (various rock domains) within the training and testing subsets is also similar.

**Table 4.** Training and Testing dataset similarity for NW (2D).

| | | Percentiles | 10% | 25% | 50% | 75% | 90% |
|:---:|:---:|:---:|:---:|:---:|:---:|:---:|:---:|
| Inputs | Training | x_train | −0.4424 | −0.1385 | 0.2293 | 0.5232 | 0.7189 |
| | | y_train | −0.4587 | −0.2334 | 0.0728 | 0.3596 | 0.5881 |
| | Testing | x_test | −0.4137 | −0.1387 | 0.2336 | 0.5017 | 0.6866 |
| | | y_test | −0.4294 | −0.2078 | 0.0742 | 0.3520 | 0.5619 |
| | Abs.diff. | x_diff | 0.0287 | 0.0003 | 0.0043 | 0.0215 | 0.0323 |
| | | y_diff | 0.0293 | 0.0256 | 0.0013 | 0.0077 | 0.0262 |
| | | | D0 | D1 | D2 | D3 | D4 | D5 |
| Outputs | Training | train_y | 5% | 44% | 36% | 3% | 10% | 2% |
| | Testing | test_y | 4% | 44% | 37% | 3% | 10% | 2% |
| | Abs.diff. | diff_y | 1% | 0% | 1% | 0% | 0% | 0% |

**Table 5.** Training and Testing dataset similarity for CEN (2D).

| | | Percentiles | | 10% | 25% | 50% | 75% | 90% |
|---|---|---|---|---|---|---|---|---|
| Inputs | Training | x_train | | −0.8543 | −0.7512 | −0.5750 | −0.2291 | 0.0781 |
| | | y_train | | −0.8257 | −0.7301 | −0.5438 | −0.2794 | −0.0304 |
| | Testing | x_test | | −0.0570 | 0.1627 | 0.4482 | 0.5789 | 0.6955 |
| | | y_test | | −0.0052 | 0.1911 | 0.4114 | 0.5669 | 0.6606 |
| | Abs.diff. | x_diff | | 0.0286 | 0.0211 | 0.0312 | 0.0503 | 0.1085 |
| | | y_diff | | 0.0518 | 0.0285 | 0.0368 | 0.0120 | 0.0349 |
| | | | D0 | D1 | D2 | D3 | D4 | D5 |
| Outputs | Training | train_y | 2% | 24% | 33% | 0% | 40% | 1% |
| | Testing | test_y | 2% | 25% | 32% | 0% | 40% | 1% |
| | Abs.diff. | diff_y | 0% | 1% | 1% | 0% | 0% | 0% |

Tables 6 and 7 show the attributes of each split for 3D modeling cases for NW and CEN deposits respectively. The tables show that the x, y and z coordinates are similar in the two subsets across several key percentiles. The distribution of various rock domains within the training and testing subsets is also similar.

**Table 6.** Training and Testing dataset similarity for NW (3D).

| | | Percentiles | | 10% | 25% | 50% | 75% | 90% |
|---|---|---|---|---|---|---|---|---|
| Inputs | Training | x_train | | −0.3845 | −0.1584 | 0.2194 | 0.5015 | 0.6665 |
| | | y_train | | −0.3960 | −0.1656 | 0.2107 | 0.4973 | 0.6620 |
| | | z_train | | −0.4902 | −0.2795 | −0.0072 | 0.2706 | 0.4817 |
| | Testing | x_test | | −0.4872 | −0.2732 | −0.0029 | 0.2668 | 0.4917 |
| | | y_test | | −0.5088 | −0.1187 | 0.3291 | 0.5458 | 0.6554 |
| | | z_test | | −0.5088 | −0.1043 | 0.3291 | 0.5458 | 0.6552 |
| | Abs.diff. | x_diff | | 0.0115 | 0.0072 | 0.0087 | 0.0042 | 0.0045 |
| | | y_diff | | 0.0029 | 0.0063 | 0.0044 | 0.0038 | 0.0099 |
| | | z_diff | | 0.0000 | 0.0144 | 0.0000 | 0.0000 | 0.0002 |
| | | | D0 | D1 | D2 | D3 | D4 | D5 |
| Outputs | Training | train_y | 5% | 38% | 44% | 3% | 9% | 1% |
| | Testing | test_y | 5% | 37% | 45% | 3% | 9% | 1% |
| | Abs.diff. | diff_y | 0% | 0% | 1% | 0% | 0% | 0% |

**Table 7.** Training and Testing dataset similarity for CEN (3D).

| | | Percentiles | | 10% | 25% | 50% | 75% | 90% |
|---|---|---|---|---|---|---|---|---|
| Inputs | Train | x_train | | −0.8710 | −0.7552 | −0.5954 | −0.2883 | −0.0281 |
| | | y_train | | −0.8721 | −0.7614 | −0.5865 | −0.2897 | −0.0511 |
| | | z_train | | −0.0244 | 0.2053 | 0.4436 | 0.5897 | 0.7225 |
| | Test | x_test | | −0.0244 | 0.2011 | 0.4378 | 0.5842 | 0.6993 |
| | | y_test | | −0.2059 | 0.0800 | 0.3976 | 0.6199 | 0.7629 |
| | | z_test | | −0.1900 | 0.0958 | 0.4135 | 0.6313 | 0.7629 |
| | Abs.diff. | x_diff | | 0.0011 | 0.0062 | 0.0089 | 0.0014 | 0.0230 |
| | | y_diff | | 0.0000 | 0.0041 | 0.0058 | 0.0055 | 0.0232 |
| | | z_diff | | 0.0159 | 0.0159 | 0.0159 | 0.0113 | 0.0000 |
| | | | D0 | D1 | D2 | D3 | D4 | D5 |
| Outputs | Train | train_y | 2% | 25% | 35% | 1% | 36% | 1% |
| | Test | test_y | 2% | 25% | 35% | 0% | 36% | 1% |
| | Abs.diff. | diff_y | 0% | 0% | 0% | 0% | 0% | 0% |

*3.2. Modeling Set Up*

As stated earlier, the deposits were modeled in 2D and 3D to predict rock domains based on coordinates. To allow modeling in 2D, the deposits were broken into 5 m thick slices. A slice is a 5 m thick interval of strata. Modeling was carried out for each slice

individually. Only slices between certain elevations were modeled to stay relevant to current mining. For the NW deposit (Table 8), a thickness of 700 m was modeled. This is equivalent to 46 15 m benches, with one additional 10-m bench. Benches are normally 15 m high and thus each bench contains three horizontal slices. Elevation ranges are described with the upper elevation specified before the lower elevation. Thus, the range is described as "1550–855", even though "855–1550" seems more intuitive to read.

**Table 8.** The data specification for the 2D method.

| Deposits | NW | CEN |
|---|---|---|
| From—to, m | 1550–855 | 1455–975 |
| Depth, m | 700 | 480 |
| Number of benches | 46 | 32 |
| Number of horizontal slices | 140 | 96 |

Slicing was not needed for 3D modeling. The modeling extents were matched with 2D modeling to allow comparisons. Table 9 gives the details of data selection for 3D modeling.

**Table 9.** The data specification for the 3D method.

| Deposits | NW | CEN |
|---|---|---|
| From—to, m | 1550–855 | 1460–915 |
| Depth, m | 700 | 550 |

## 4. ML Modeling Methods and Results

Several ML techniques were utilized in this research to predict the rock domains (output) based on coordinates (inputs). They are not described here as they are considered standard techniques now. The interested reader will find numerous resources on each of the techniques if they wish for an introduction to any of them. The public domain Scikit-learn toolkit [14] was used for implementing the techniques. For those unfamiliar with Python-based machine learning tools, Scikit-learn is a large warehouse of public domain tools that is supported by some of the largest technology companies in the world. These tools are not only used commercially by technology companies but are also popular among academic researchers.

ML techniques typically require numerous hyperparameters. These are discussed when results are presented. Unless specified, the techniques were applied using default settings.

### 4.1. 2D Modeling Results

4.1.1. Impact of Hyperparameters and Modeling Method

There is no succinct way to show the results of the numerous models that were developed. For 2D modeling, there were 140 slices. Several models had to be developed for each slice. Therefore, figures were generated and presented to demonstrate the most important conclusions.

Figure 1 shows the results from the application of multilayer perceptron (MLP) or neural network in the 2D framework in the NW deposit. This was implemented using Sci-kit's MLPClassifier() tool [15]. The figure shows the scores for various slices (indicated by their elevation). Scores simply mean the proportion of test cases that were accurately predicted. The activation function used was "softmax", and the optimizer was "LBGFS". In all the MLP modeling conducted in this paper, the numbers of neurons and layers were varied. However, the fine-tuning of the layers and neurons is not reported in this paper, as that is a trivial exercise and not particularly insightful. Instead, the role of the L2 regularization parameter ($\alpha$), which is one way of generalizing MLPs, was explored and reported on. It varied from 0.1 to 10. Results are presented to show the impact of this hyperparameter.

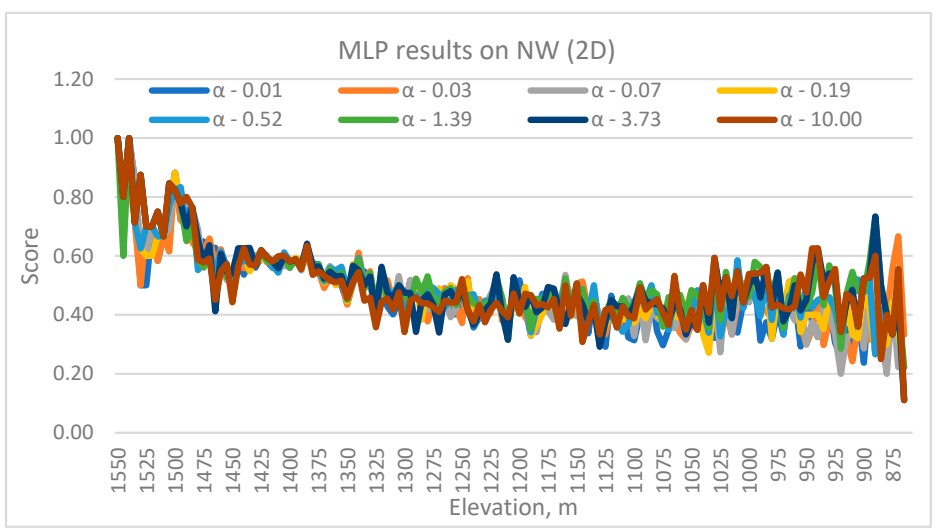

**Figure 1.** Variation in MLP prediction performance with elevation and $\alpha$ in NW.

Two things are immediately apparent from Figure 1. One, the MLP classifier performed much better at 1550–1500 elevations than everywhere else. Generally speaking, the performance appeared to drop with depth. The average score was 0.76 in the 1550–1500 m elevation range. Second, $\alpha$ impacts performance but there is no clarity on any specific value. The best performance is 1, implying a 100% prediction accuracy. However, that is not particularly impressive as there were just four samples in the test case at that elevation since these elevations represent a hill.

Figure 2 shows the effect of $\alpha$ on MLP models for NW deposit. Between Figures 1 and 2, it appears that there is not a single $\alpha$ that is perfect. Different $\alpha$ work best at different elevations. The median score for all the $\alpha$'s is 0.47. This could be classified as fairly poor performance.

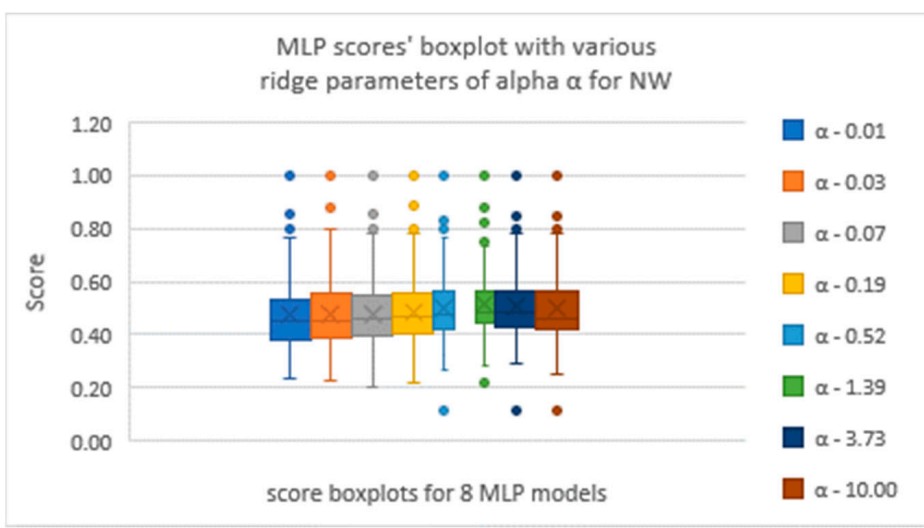

**Figure 2.** Box plots of results in various $\alpha$ on MLP performance in NW.

Figures 3 and 4 show the MLP performance in the CEN deposit. The overall trend is somewhat different with respect to depth than for the NW deposit. The deeper benches have done as well as the upper elevations on average, though the effect of $\alpha$ was highly erratic in the deeper elevations. The median score, 0.43, is a bit lower than the NW deposit.

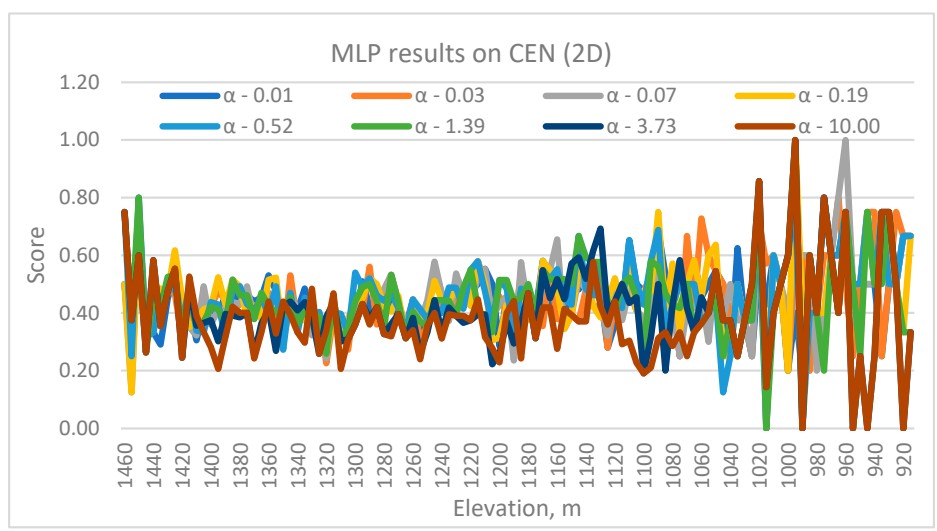

**Figure 3.** Variation in MLP prediction performance with elevation and α in CEN.

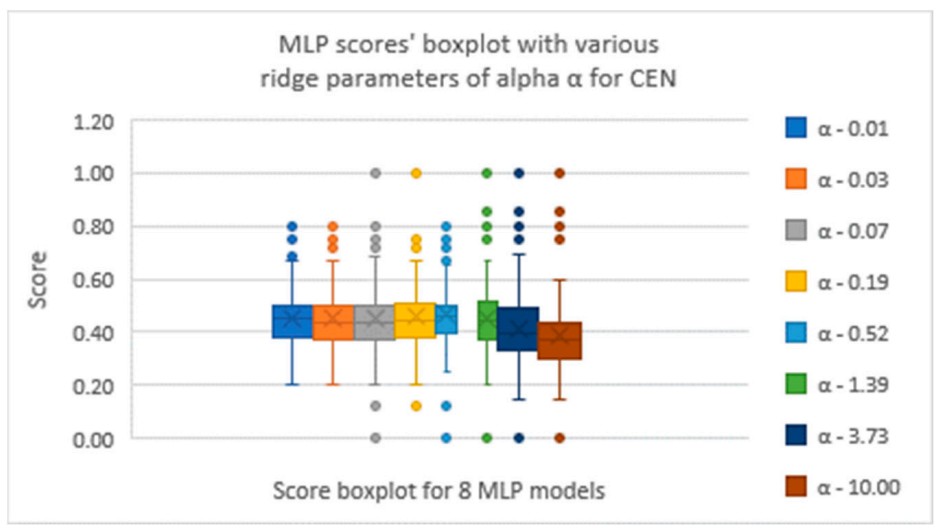

**Figure 4.** Box plots of results in various α on MLP performance in CEN.

Another technique that was applied was the k-nearest neighbor (KNN). In this technique [16], the estimate is based on the k closest neighbors. The weight assigned to a neighbor is inversely related to the distance to the point being estimated. The default setting, Euclidean distance, was used for the definition of distance.

Many different neighborhood sizes, from 3 to 11, were tried (Figure 5). The median performance is 0.46 for the NW mining area and 0.42 for the CEN area. The size of the neighborhood did not make much of a difference to the CEN deposit. For the NW deposit, neighborhood sizes 7 to 11 seemed to perform a little bit better than neighborhood sizes 3 and 5. Regardless, the performance should be classified as poor.

Random forests (RF) were also tried [17], with Maximum Tree Depths (MTD) of 6, 8, 11 and 14 (Figure 6). The performance remained about the same barring minor differences. Continuing with the general trend, RF performed a little bit better in the NW area than in the CEN area, but once again, the overall median performance is poor (0.47 for NW and 0.44 for CEN).

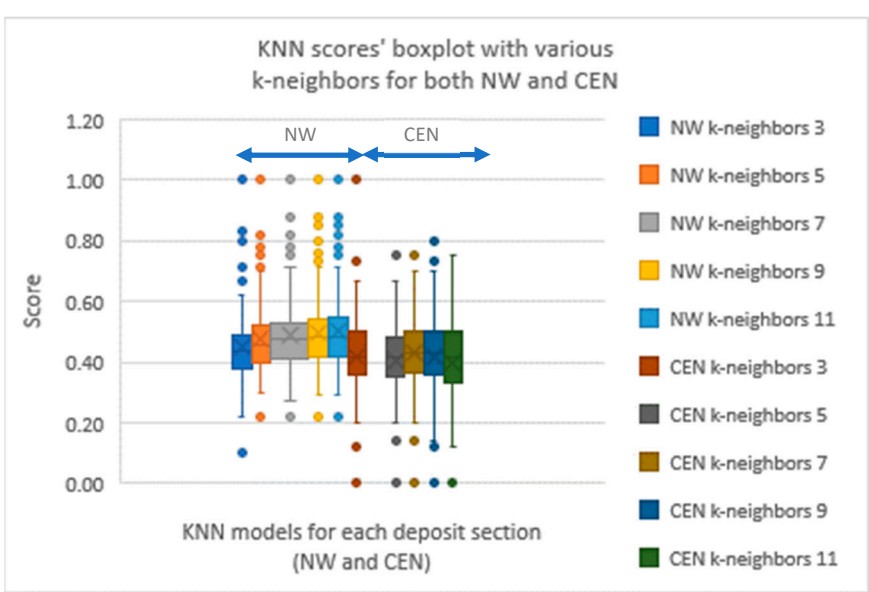

**Figure 5.** Box plots of results in various k-numbers on KNN performance in both NW and CEN.

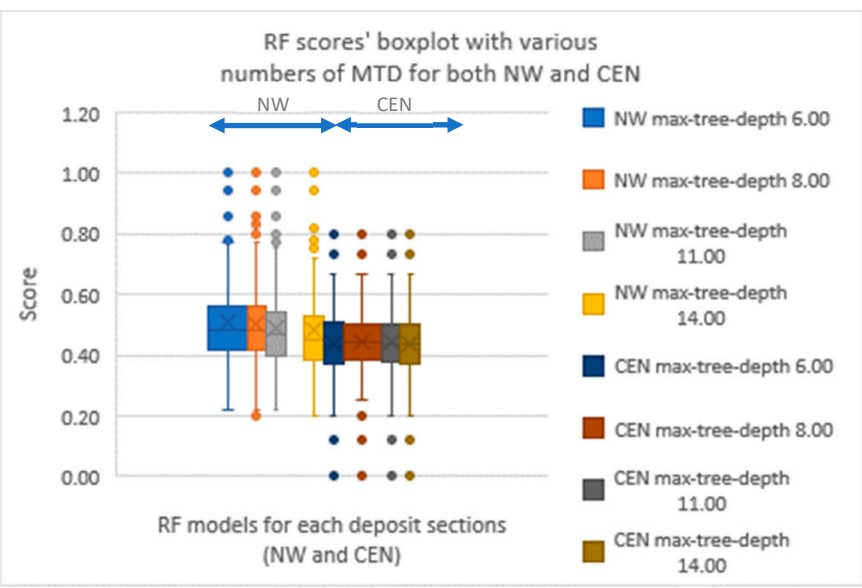

**Figure 6.** Box plots of results in various MTD on RF performance in both NW and CEN.

### 4.1.2. Performance in Different Rock Domains

Tables 10 and 11 show the confusion matrix summarizing the best performances of 2D modeling (RF) in the NW and CEN deposits, respectively. The test set had 569 samples from the D0 domain. Of them, only one was correctly predicted as D0. This is shown as 0.00 in Table 10 (row D0, column D0). Most D0 samples were predicted as D1 (203) or D2 (357). This is shown as 0.36 (36%) and 0.63 (63%) in the table. Though terrible, this performance is understandable as D0 constituted only 4.7% of the data in NW deposit. The performance of each rock domain is presented as the diagonal element (shaded). Performance was similarly bad for D0 in CEN. The performance in D1 was better, with 56% of the D1 samples in the NW deposit being predicted as D1. Performance in D2 was similar, with 66% of the D2 samples in the NW deposit being predicted as D2. From the tables, one pattern that emerges is that the model underperformed significantly in predicting a domain if that domain did not constitute a certain proportion of the data. Thus, performance in D1 and D2 was much higher for the NW deposit than in D0, D3 and D5. In the CEN

deposit, performance was much higher in D1, D2 and D4 than in D0 and D5. The second pattern that emerges is that when domains D0, D3, D4 and D5 are predicted wrong, they are usually predicted as D1 or D2 (since these two domains constitute almost 80% of the data). For example, 47% of D3 samples are predicted as D1 in NW deposit and 50% are predicted as D2.

**Table 10.** Confusion matrix for 2D RF modeling for NW.

| Rock Type | | Predicted As | | | | | |
|---|---|---|---|---|---|---|---|
| | | **D0** | **D1** | **D2** | **D3** | **D4** | **D5** |
| Rock Domain | D0 | 0.00 | 0.36 | 0.63 | 0.00 | 0.01 | 0.00 |
| | D1 | 0.00 | 0.56 | 0.43 | 0.00 | 0.00 | 0.00 |
| | D2 | 0.00 | 0.33 | 0.66 | 0.00 | 0.01 | 0.00 |
| | D3 | 0.00 | 0.47 | 0.50 | 0.03 | 0.00 | 0.00 |
| | D4 | 0.00 | 0.51 | 0.45 | 0.00 | 0.03 | 0.00 |
| | D5 | 0.01 | 0.25 | 0.56 | 0.02 | 0.02 | 0.15 |

**Table 11.** Confusion matrix for 2D RF modeling for CEN.

| Rock Type | | Predicted As | | | | | |
|---|---|---|---|---|---|---|---|
| | | **D0** | **D1** | **D2** | **D3** | **D4** | **D5** |
| Rock Domain | D0 | 0.03 | 0.21 | 0.48 | 0.00 | 0.28 | 0.00 |
| | D1 | 0.01 | 0.38 | 0.30 | 0.00 | 0.31 | 0.00 |
| | D2 | 0.01 | 0.19 | 0.44 | 0.00 | 0.36 | 0.00 |
| | D3 | 0.06 | 0.31 | 0.31 | 0.00 | 0.31 | 0.00 |
| | D4 | 0.00 | 0.19 | 0.30 | 0.00 | 0.51 | 0.00 |
| | D5 | 0.00 | 0.24 | 0.21 | 0.00 | 0.41 | 0.15 |

For brevity's sake, the confusion matrices for MLP and KNN are not shown as conclusions are similar.

### 4.2. 3D Modeling Results

#### 4.2.1. Impact of Methods and Hyperparameters in 3D Modeling

In 3D modeling, the entire space could be modeled at once, and thus, results could be presented much more elegantly. In 2D, there were over 100 slices in each area which complicated the presentation of results. Figure 7 shows the results for MLP, KNN and RF for NW and CEN areas. It is clear that RF and KNN performed much better than MLP and much better than in the 2D case. MLP also performed better than the 2D case. Performance in the CEN deposit reached 0.95 and was better than in the NW deposit, where the highest score was 0.88. RF outperformed KNN in both deposits by a small margin. The best tree depth was 24, and the best neighborhood size was 3. The smaller neighborhood size makes sense given the tree depth. As trees get deeper, the bottom leaves would not have that many samples.

#### 4.2.2. Performance on Different Rock Domains

Tables 12 and 13 show the confusion matrix for the 3D modeling performance of RF in NW and CEN deposits, respectively. Table 12 show that D1 and D2 were predicted very well in NW deposit, with 92% of D1 predicted as D1 and 94% of D2 predicted as D2. D4 was also predicted well (85%) despite constituting only 9.4% of the samples. In CEN deposit, performance was high for D1 (94%), D2 (96%) and D4 (98%). As with 2D modeling, performance was higher for rock domains that constituted a certain minimum proportion of the data. Also, just like in the 2D case, when samples were misclassified they were likely to be classified as D1 or D2.

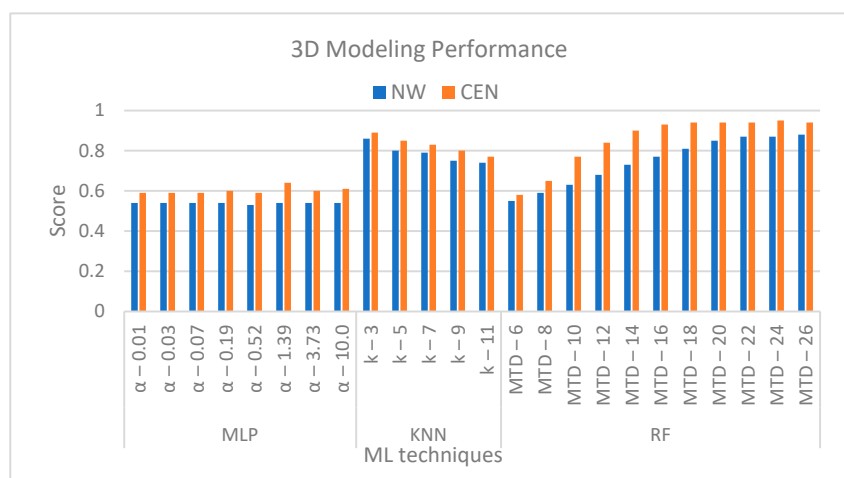

**Figure 7.** Variation in ML prediction performance in both NW and CEN.

**Table 12.** Confusion matrix for 3D RF modeling for NW.

| Rock Type | | Predicted As | | | | | |
|---|---|---|---|---|---|---|---|
| | | **D0** | **D1** | **D2** | **D3** | **D4** | **D5** |
| Rock Domain | D0 | 0.31 | 0.30 | 0.37 | 0.01 | 0.01 | 0.01 |
| | D1 | 0.01 | 0.92 | 0.07 | 0.00 | 0.00 | 0.00 |
| | D2 | 0.01 | 0.04 | 0.94 | 0.00 | 0.00 | 0.00 |
| | D3 | 0.02 | 0.33 | 0.24 | 0.41 | 0.00 | 0.01 |
| | D4 | 0.00 | 0.07 | 0.07 | 0.00 | 0.85 | 0.00 |
| | D5 | 0.01 | 0.16 | 0.24 | 0.01 | 0.02 | 0.58 |

**Table 13.** Confusion matrix for 3D RF modeling for CEN.

| Rock Type | | Predicted As | | | | | |
|---|---|---|---|---|---|---|---|
| | | **D0** | **D1** | **D2** | **D3** | **D4** | **D5** |
| Rock Domain | D0 | 0.39 | 0.23 | 0.39 | 0.00 | 0.00 | 0.00 |
| | D1 | 0.00 | 0.94 | 0.03 | 0.00 | 0.02 | 0.00 |
| | D2 | 0.01 | 0.01 | 0.96 | 0.00 | 0.01 | 0.00 |
| | D3 | 0.00 | 0.31 | 0.31 | 0.38 | 0.00 | 0.00 |
| | D4 | 0.00 | 0.01 | 0.00 | 0.00 | 0.98 | 0.00 |
| | D5 | 0.00 | 0.03 | 0.18 | 0.00 | 0.12 | 0.68 |

Omni-directional indicator semi-variograms (ISV) were computed for the rock domains to see if they could help explain the vastly different ML performance in the different rock domains. The ranges of the ISVs are reported in Table 14. Except for D1 in CEN, the domains that were predicted well had much longer ranges than domains that were not predicted well. It appears that the predictability of the domain depended not only on what proportions it contributed to the total sample size but also on the ISV range. This makes sense as a longer range implies more continuity. The sills were not reported in Table 14 as sills depend on the total number of samples and are thus not comparable across domains.

**Table 14.** Indicator semi-variogram ranges in meters for the various rock domains in NW and CEN.

| | NW | CEN |
|---|---|---|
| D0 | 50 | 65 |
| D1 | 150 | 80 |
| D2 | 120 | 140 |
| D3 | 75 | 75 |
| D4 | 120 | 130 |
| D5 | 50 | 100 |

## 5. Discussion

The two mining areas, NW and CEN, were modeled in 2D and 3D to better understand the distribution of various rock types. This resulted in very different ML performances. Due to low prediction accuracy, rock domains cannot be properly predicted with 2D models. The performance of 3D models was much higher, with KNN and RF significantly outperforming MLP. Overall, RF did the best. 3D ML performance was better in CEN deposit than in NW deposit.

The domains that were predicted the best in both deposits were D1, D2 and D4. It is interesting that D4 was predicted so well in the NW deposit despite constituting only 9% of the data, compared to 25% and 35% for D1 and D2 respectively. The fact that D4 ISV in NW had a range comparable to D1 and D2 may explain the better modeling of the domain.

The domains D0, D3 and D5 could not be predicted well at all, with none exceeding 5% of the data. The domains D3 and D5 constituted less than 3% of the data. The prediction performance for a domain appears to depend not only on how much it contributed to the total sample size but also on the ISV range. This is not entirely surprising, as according to [10], a shorter range increases the estimation variance.

It is not clear why MLP did not do as well as RF or KNN. There was also no real insight into the proper L2 regularization hyperparameter. It was not surprising RF did well since previous work had demonstrated its high performance in predicting the granodiorite rock type [1]. What was surprising was that the neighborhood size of 3 outperformed larger neighborhood sizes in the KNN method. This appears related to the fact that RF worked best when trees were deep. In deep trees, predictions are made from a smaller number of samples.

The high performance in small a neighborhood size, or with deep trees, appears to be in conflict with performance being higher in domains with a longer range. When the range is longer you would expect larger neighborhoods to be effective. However, there is no conflict. When the range is short the variability climbs very quickly. Thus, the three closest neighbors that are selected for making a prediction could be quite different from the value being predicted, whereas three neighbors that are selected when the range is high, are likely to be similar to the value being predicted. Thus, performance should be expected to be better when the range is higher regardless of ideal neighborhood sizes. This does not mean a longer range would automatically yield larger ideal neighborhoods. Since no rock domain vastly dominates the data, omni-directional increase in neighborhood size simply introduces variability. If the KNN and RF methods could be modified to honor directional relationships (like what is done in the inverse distance squared method in geostatistics when a directional search radius is applied), neighborhood sizes and tree depth would not be fixed but depend on anisotropy. Examining this issue would be a great next step for future research.

## 6. Conclusions

ML was applied to the drillhole database at the large Erdenet copper mine in Mongolia to determine if rock domains could be predicted with any reliability. Two of the currently mined deposits, NW and CEN, were the focus of the research. To understand the spatial nature of the deposits, modeling was done in both 2D and 3D. Modeling in 2D vastly underperformed modeling in 3D. Given the poor performance in 2D, the rest of the conclusions are drawn only from the results of 3D modeling.

The 3D modeling performance was excellent for the two most important and prominent rock domains, D1 and D2. Over 90% of the two domains, D1 and D2, were predicted correctly. When misclassified, D1 was likely to be classified as D2. Similarly, when misclassified, D2 was likely to be classified as D1. Since D1 and D2 constituted about 80% of the samples, most domains were misclassified as D1 or D2.

ML performance was quite high for D4 even though it constituted only 15% of the samples between the two deposits. This may be explained by the longer range of its ISV. The results in the various rock domains seemed to indicate that predictability depended on

two things: proportion of samples from the domain in question, and the range of the ISV for that domain. The longer range appeared to positively impact ML performance.

MLP did not perform as well as KNN and RF. The hyperparameters that worked best for RF explained the hyperparameters that worked best for KNN. Performance was high when the number of samples used to predict rock domain was low. In the case of KNN, the neighborhood size of 3 worked best. Increasing neighborhood sizes or lowering tree depth, both of which have a similar effect, did not improve predictability as involving more samples in making a prediction simply introduced more variability. This is because each domain contributed to only a small proportion of samples.

In summary, three conclusions can be drawn directly. One, the rock domain could be modeled in 3D in Erdenet using ML with high prediction accuracy. Two, 2D modeling was not successful. Three, variogram parameters were indeed insightful about ML performance.

More generally, there are two important conclusions. This paper demonstrated that ML can be quite beneficial in assisting mines with a day-to-day understanding of the rock domains. Additionally, when variogram modeling is done, the range and the sill can be insightful to gauge the performance that is expected from ML methods.

**Author Contributions:** Conceptualization, N.S., R.G.; data curation, N.S.; formal analysis, N.S., R.G.; funding acquisition, R.G.; investigation, N.S., R.G.; methodology, N.S., R.G.; validation, N.S., R.G.; visualization, N.S., R.G.; writing—original draft, N.S., R.G.; writing—review and editing, N.S., R.G. All authors have read and agreed to the published version of the manuscript.

**Funding:** This work was performed as part of an agreement between the University of Utah and Erdenet Mining Corporation LLC.

**Data Availability Statement:** Not applicable.

**Acknowledgments:** The help of the geologists and engineers at EMC and professors from the University of Utah is gratefully acknowledged.

**Conflicts of Interest:** The authors declare no conflict of interest. The funders had no role in the design of the study; in the collection, analyses, or interpretation of data; in the writing of the manuscript, or in the decision to publish the results.

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
