# Peer review of "Gaining Insight from Semi-Variograms into Machine Learning Performance of Rock Domains at a Copper Mine"

_minerals, doi:10.3390/min12091062_

Round 1

Reviewer 1 Report

Manuscript Number:   minerals-1862477

The manuscript has some shortcomings which need to be improved prior to its publication. My recommendation is that the article needs Major Revisions before it can be considered for publication.

1.      Abstract: The abstract is a bit generic. Please add some more information regarding your results. It should be improved in a quantitative way.

2.      Introduction is generalized. I would recommend following recent research articles to reconstruct with extensive literature review.

https://doi.org/10.3390/rs12172833

3.      Author should write more about study area.

4.      Methodology section is weakly written. So, my suggestion is to reconstruct it.

 https://doi.org/10.3390/rs12213568

5.      The author may write discussion section more elaborately and compare it with recent publications to justify your findings.

6.      The author should include one more section of materials and methods, under which author can write materials and method section.

7.      In conclusion section, you must mention the implications of your research and how it makes a footprint in scientific research. Try to incorporate your work to global interest how this research has worldwide importance. It will be interesting for the readers.

8.      Reference: Re-check the whole reference just to make sure you have added all the references that you cited in your manuscript.

9.      Apart from this the quality of the overall paper is very good. I prefer this article with acceptable with major modifications.

Author Response

Thank you for reviewing our paper.

Here is attached the file response for your review.

Warm Regards,

Authors

Reviewer 2 Report

The manuscript deals with modelling rock properties of a large copper deposit where a mining site has been developed, based on a  drill hole database, using machine learning techniques. The research issue is significant considering the innovative machine learning techniques, which constitute a very current and continuously emerging sector in the scientific community. Rock variations assessment in mining sites is complex due to the time-consuming, costly, and subjective processes. Machine learning techniques aim at solving such issues. The authors tried assessing drill hole data to model the rock domain properties.

However, some important issues need to be improved. Although the research issue is of high importance and interest, it seems that the rock domain modelling needs to be further explained considering the initial target of the research.

Furthermore:

-          From an engineering point of view, the practical considerations of the research should be further discussed.

-          Compared to other previous works, the original contribution of the work needs to be clearly presented.

-          A more critical review of the literature is needed concerning the research questions. Unfortunately, the literature provided in the paper is relatively poor regarding the research issue.

-          Very short sentences exist throughout the manuscript, and the manuscript is written in not such a formal style. The whole text needs to be revised regarding the way of writing.

-          The structure of the manuscript needs to be improved. The sections are not clearly separated. In particular, the methods that were used are not clearly presented.

-          The title refers to semi-variograms. However, no semi-variogram appeared or was adequately discussed in the manuscript.

-          A flowchart with the methodology that was followed could help the readers.

-           It is suggested to avoid using exact words of the title as keywords.

Abstract

General note: The abstract needs to be improved and also be more informative regarding the main objective, the methods applied, and the paper's main results. Now it does not highlight the contribution of the work.

[Line 9] "Previously…" When previously? An explanation is needed here.

[Line 11] "As a result of the optimistic results…" Which are the optimistic results? Also, the main results need to be mentioned in the abstract.

[Line 14 & 17] "2D modeling performance was poor" There is a sentence repetition.

[Lines 17-19] Three different sentences refer to "Performance" Please improve the sentences.

1.    Introduction

General note: The introduction needs to be more critical, especially regarding the methods used in the international literature. In addition, the research questions and the main objectives of the manuscript need to be added in this section.

[Line 30] "Machine learning (ML) is increasingly…" This part of the sentence has also been mentioned previously.  

[Line 31] "…very manual…"? It needs to be replaced by a more suitable expression.

[Line 41-42] "…when rock type… operating bench" It seems that this sentence is not comprehensive.

[Line 47] "Given the optimistic results…' Please clarify what these results are and why they are characterized as optimistic".

[Line 57] "…may be like." The sentence needs to be improved. Also, the "others" should be replaced by a more formal expression.

[Line 59] "In that, this paper is a first." Please improve this sentence.

  1. Geology and Drill Hole Data

General note: A map of the study area and the location of the mine deposits could be helpful for the readers. Furthermore, the domains in some parts are written in capital letters (e.g. "D0"), whereas, in Table 1 and some other parts of the text, they are written in small letters (e.g. "d0"). Please follow one style.

 [Lines 67-68] The sentence makes no sense. Please improve it.

[Line 82] Please explain what the "unknown" shows.

[Line 91] Table 1: The sums of the 5th column are not equal to 100, but 99.7. Please check them. Also, the same decimal digits in all columns must be used. 

 3. Machine Learning Setup

General notes: Please explain why the specific percentiles presented in the Tables were selected. Why are they characterized as "key percentiles"? What is the practical usefulness of presenting the percentages in the Tables? Furthermore,  the maps of 3D or 2D modelling approaches need to be improved to clarify the results.

[Line 96] Which coordinates? The sentence introduces the issue very sudden. A further description is needed.

[Line 101] Table 1 needs to be further described in the text.

 [Line 116] "The deposits were modelled in 2D and 3D…" This sentence is a repetition here.

[Line 120] "…forty-six 15 m benches…" Please improve it and explain. Also, the "forty-six" could be written as 46.

[Lines 132-133] The sentence needs to be improved.

.     4. ML Modeling and Results

 [Line 145] What slices? An explanation is needed here.

[Line 158 & 177] The lines of the diagrams in Figures 1 and 3 should be thinner for all the results to be better understood.

[Line 160] Figure 1 instead of Figure 7?

  1. Discussion

[Line 259] "It was very clear…models" Why was it clear? A further explanation is needed.

  1. Conclusions

General notes: The manuscript mainly focuses on the machine learning techniques that were used and less on the rock domains modelling. In addition, in the "Conclusions" sections, a comparison of the machine learning techniques is discussed and not the initial aim of the manuscript. In this framework, the conclusions must be improved by focusing on the original contribution of the research.

Author Response

Thank you for reviewing our paper.

Here is the attached response file for your review.

Warm Regards,

Authors

Round 2

Reviewer 1 Report

This article can be accepted in its present form for publication.

Reviewer 2 Report

The manuscript has been significantly improved in the revised version, and most of my comments have been addressed. Although further improvement could be made in order to increase the readability of the work, I believe that the manuscript in the present form warrants publication.

A clarification on my comment referring to [Lines 67-68] of the initial version of the manuscript. Of course, the sentence is informative to a geologist, but what about the expression “ the surface mine mines the deposit”?